# Outcomes of Chronic Phase Chronic Myeloid Leukemia after Treatment with Multiple Tyrosine Kinase Inhibitors

**DOI:** 10.3390/jcm9051542

**Published:** 2020-05-20

**Authors:** Jee Hyun Kong, Elliott F. Winton, Leonard T. Heffner, Manila Gaddh, Brittany Hill, Jessica Neely, Angela Hatcher, Meena Joseph, Martha Arellano, Fuad El-Rassi, Audrey Kim, Jean Hanna Khoury, Vamsi K. Kota

**Affiliations:** 1Department of Hematology Oncology, Division of Internal Medicine, Wonju Severance Christian Hospital, Yonsei College of Medicine, Wonju 26426, Korea; kkongg@yonsei.ac.kr; 2Department of Hematology and Medical Oncology, Winship Cancer Institute of Emory University, Atlanta, GA 30322, USA; Ewinton@emory.edu (E.F.W.); lheffner@emory.edu (L.T.H.); manila.gaddh@emory.edu (M.G.); brittany.hill@emoryhealthcare.org (B.H.); jessica.neely@emoryhealthcare.org (J.N.); angela.hatcher@emoryhealthcare.org (A.H.); meena.joseph@emoryhealthcare.org (M.J.); marella@emory.edu (M.A.); fuad.elrassi@emoryhealthcare.org (F.E.-R.); audreyskim@gmail.com (A.K.); hkhoury@emory.edu (J.H.K.); 3Section of Hematology and Oncology, Georgia Cancer Center at Augusta University, Augusta, GA 30912, USA

**Keywords:** chronic myeloid leukemia, chronic phase, tyrosine kinase inhibitor, survival rates, treatment-free remission

## Abstract

We sought to evaluate the outcomes of chronic phase (CP) chronic myeloid leukemia (CML) in an era where five tyrosine kinase inhibitors (TKIs) are commercially available for the treatment of CML. Records of patients diagnosed with CP CML, treated with TKIs and referred to our center were reviewed. Between January 2005 and April 2016, 206 patients were followed for a median of 48.8 (1.4–190.1) months. A total of 76 (37%) patients received one TKI, 73 (35%) received two TKIs and 57 (28%) were exposed to >3 TKIs (3 TKIs, *n* = 33; 4 TKIs, *n* = 17; 5 TKIs, *n* = 7). Nineteen (9.2%) patients progressed to advanced phases of CML (accelerated phase, *n* = 6; myeloid blastic phase, *n* = 4; lymphoid blastic phase, *n* = 9). One third (*n* = 69) achieved complete molecular response (CMR) at first-line treatment. An additional 55 patients achieved CMR after second-line treatment. Twenty-five patients (12.1%) attempted TKI discontinuation and 14 (6.8%) stopped TKIs for a median of 6.3 months (range 1–53.4). The 10-year progression-free survival and overall survival (OS) rates were 81% and 87%, respectively. OS after 10-years, based on TKI exposure, was 100% (1 TKI), 82% (2 TKIs), 87% (3 TKIs), 75% (4 TKIs) and 55% (5 TKIs). The best OS was observed in patients tolerating and responding to first line TKI, but multiple TKIs led patients to gain treatment-free remission.

## 1. Introduction

Chronic myeloid leukemia (CML) is a clonal myeloproliferative stem cell disorder characterized by the presence of a signature hybrid oncogene, the *BCR-ABL1* [1]. The *ABL1* gene encodes non-receptor tyrosine kinases that become deregulated and constitutively active by the juxtaposition of BCR. BCR-ABL plays a central role in controlling the downstream pathways involved in cell proliferation, the regulation of cellular adhesion and apoptosis [1]. The understanding of the pathophysiology of CML led to the development of drugs that specifically target the tyrosine kinase activity of BCR-ABL. Currently, there are five tyrosine kinase inhibitors (TKIs) approved for the treatment of CML: Imatinib (IM, Gleevec^®^, Novartis Oncology, East Hanover, NJ, USA), dasatinib (DAS, Sprycel^®^, Bristol-Meyers Squibb Company, Princeton, NJ, USA), nilotinib (NIL, Tasigna^®^, Novartis Oncology, East Hanover, NJ, USA), bosutinib (BOS, Bosulif^®^, Pfizer, New York, NY, USA) for both first and second-line therapy, and ponatinib (PON, Iclusig^®^, Ariad Pharmaceuticals, Cambridge, MA, USA) for patients with the T315I mutation or for whom no other TKI is indicated. Indeed, TKI treatment for patients with CML is ranked as one of the greatest medical success stories of the past 30 years, which translates into life spans indistinguishable from those of individuals without leukemia [2]. Even though a large proportion of patients exhibit a prolonged molecular response, a non-negligible number of these patients require an alternative to TKI treatment due to resistance as well as intolerance or toxicity [3]. However, the outcome of patients requiring multiple TKI treatments in chronic phase (CP) CML has not been well established. Previous reports have included patients treated with interferon or chemotherapy prior to TKI [4,5,6,7,8], advanced phase [4,9,10,11] or *ABL* kinase mutation [4,10]. Furthermore, outcomes have been reported only after second- [8,12,13], third- [7,14], or fourth-line [14] TKI treatments. In addition to survival, discontinuing TKI for treatment-free remission has become another goal for CML treatment [15], and the incidence of TKI discontinuation is not known other than in first-line TKI treatment. Thus, we sought to evaluate the outcomes of CP CML in an era where several TKIs are available.

## 2. Materials and Methods

We reviewed the records of 284 patients diagnosed with CML, who were referred to the Winship Cancer Institute of Emory University between January 2005 and April 2016. Epidemiological information, CML-specific characteristics including blood and bone marrow test results, treatment, response to therapy, and reason for TKI switching, were extracted. We excluded accelerated phase (AP, *n* = 23) or blast phase (BP, *n* = 12) patients. Patients treated with therapies other than TKIs as first-line (*n* = 11), those with a long interval between TKI and diagnosis (*n* = 2), and those who did not have initial diagnosis or treatment data (*n* = 29) were excluded.

CP, AP, and BP were defined according to standard criteria [16,17]. The European Leukemia Net (ELN) criteria were used to define hematological and molecular responses [18]. Qualitative real-time polymerase chain reaction (RT-PCR) results for *BCR-ABL1* have been available since 2010. Major molecular response (MMR) is when the ratio of BCR/ABL1 transcript to ABL1 transcript is < 0.1%, and complete molecular response (CMR) is BCR/ABL1 that is not detected in quantitative or qualitative RT-PCR with a sensitivity of at least 0.0063% (international scale). Bone marrow or blood fluorescent in situ hybridization (FISH) was performed 12 months after TKI treatment in 81 patients only; hence, we did not collect the cytogenetic response outcomes. The line of treatment was defined independently from the number of exposures to TKIs. For example, if a patient used IM in the frontline setting, then used DAS as second-line treatment, and used IM again (after DAS), then the last IM was classified as third-line treatment, although 2 TKIs were used. We collected the reasons for TKI change or discontinuation and divided them into 3 categories: ‘resistant’ (resistant, suboptimal response, or treatment failure), ‘intolerance’ (intolerance or adverse events), and ‘other reasons’ (insurance, clinical trial, or not available). It was not known that whether resistance to TKI was primary or secondary, because most patients were referred from local clinics.

### Statistical Analysis

Fisher’s exact test and the χ^2^ test were used to compare categorical variables, and the Mann–Whitney U test was used for continuous variables. Progression-free survival (PFS) was calculated from the date of diagnosis to the date of accelerated phase or blastic crisis, and overall survival (OS) rates were calculated from diagnosis to death, or last follow-up using the life tables and the Kaplan–Meier method. The log-rank test was used to identify factors affecting PFS or OS. Missing data were not included in this analysis. Univariate analysis for PFS and OS included the following variables: sex, age (<50 vs. ≥50 years), race, risk scores, additional chromosomal abnormalities, number of TKIs used before disease progression (for PFS) or last follow-up (for OS), first-line TKI treatment (IM vs. newer generation TKIs (NG-TKI)), *ABL* kinase mutation, and BCR/ABL^IS^ at 3, 6 and 12 months. *p* < 0.05 was considered statistically significant for all analyses. The SPSS 25.0 (Armonk, NY: IBM Corp) statistical program was used for all statistical analyses.

## 3. Results

### 3.1. Patients’ Characteristics

A total of 206 patients, with a median follow-up period of 48.8 (1.4–190.1) months after diagnosis, were included. The patients’ clinical characteristics are summarized in Table 1. The median age at diagnosis was 50 (11–88) years and the male/female ratio was 105/101. The most frequently observed race was white (51.5%), followed by African American (27.2%), Hispanic (2.9%), Asian (2.4%) other (1.5%) or unknown (14.6%). Risk scores, including Sokal, Hasford, and EUTOS [19] were calculated for 110 (53%) patients, of whom 55 (50%) were low risk in all the scoring systems. Bone marrow reports were available for 173 (84%) patients, whereas five patients did not undergo bone marrow biopsy at the initial diagnosis. After excluding Y deletion (*n* = 2) and the Philadelphia chromosome variant (*n* = 5), 12 (5.8%) patients showed additional chromosomal abnormalities (ACA), and two were major route ACAs: trisomy 8 (*n* = 1) and trisomy 19 (*n* = 1) (Table 1) [20].

All patients initiated TKI treatment after diagnosis. In addition to the five FDA-approved TKIs, rebastinib (REB) was used in two patients as part of the clinical trial at second- and fourth-line treatments. Initial TKI was maintained in 76 patients (37%), while 73 (35%) were treated with two TKIs and the remainder (*n* = 57, 28%) with >2 TKIs (Figure 1a). Eighteen patients (9%) reused the same TKI in different treatment lines. IM was the most frequently used in first-line treatment (*n* = 145, 70%). After DAS and NIL were approved for the frontline treatment of CP CML by the FDA, 113 out of 206 patients initiated TKI, and IM was selected in 52 patients, followed by DAS (*n* = 43), NIL (*n* = 13), BOS (*n* = 4) and PON (*n* = 1), respectively. In second-line treatment, DAS (*n* = 68, 52%) was used in more than half of the selected patients. NIL was the most frequently prescribed TKI in third-line (*n* = 21, 32%) and PON was used in fourth- and fifth-line treatments (4th; *n* = 11, 37%, and 5th; *n* = 4, 40%) (Table 1). TKIs were changed due to resistance in 50%, intolerance in 43%, and other reasons in 7% of patients.

Nivolumab, blinatumomab, and chemotherapy (Hyper-CVAD (cyclophosphamide, vincristine, doxorubicin, and dexamethasone), decitabine, and high-dose cytarabine) were administered after TKI or concomitant to TKI in three (1.5%), one (0.5%), and 12 (5.8%) patients, respectively. Eight (3.9%) patients underwent allogeneic hematopoietic stem cell transplantation (HSCT), and one (0.5) underwent autologous HSCT. The reasons for allogeneic HSCT were disease progression to AP (*n* = 1) and lymphoid BP (LBP) (*n* = 3), TKI failure (*n* = 2), T315I mutation (*n* = 1) or persistent neutropenia after Imatinib (*n* = 1). The autologous HSCT was performed due to recurrent leukemic meningitis (*n* = 1). The *ABL* domain mutation was tested in 72 (34%) patients, and mutations were detected in 25 (35%). Nine patients had more than one mutation, and T315I was detected in 9 (36%) of them.

### 3.2. Clinical Outcomes

At the median of 39.0 (3.8–119.8) months after diagnosis, 19 (9.2%) patients progressed to advanced phases of CML (AP, *n* = 6; myeloid BP (MBP), *n* = 4; LBP, *n* = 9). Fifteen were treated with a median of 1.5 (1–3) more TKIs after progression. The estimated cumulative PFS rates at five and 10 years for all patients were 92% and 81%, respectively (Figure 1a). Among the 19 patients who progressed, nine events occurred during frontline treatment and the number of TKIs used before disease progression did not make any difference overall (*p* = 0.051, Figure 1a). However, the number of TKIs used was significant for PFS, with fifth-line TKI-treated patients having a lower PFS rate compared to the patients who received 1–3 TKIs (one TKI vs. five TKIs, *p* = 0.032; two TKIs vs. five TKIs, *p* = 0.016; three TKIs vs. five TKIs, *p* = 0.014) (Figure 1a).

Eleven patients (5.3%) died due to a refractory disease (*n* = 6), complications of allogeneic HSCT (*n* = 4), and acute myocardial infarction (*n* = 1). The estimated cumulative OS rates at five and 10 years were 94% and 87%, respectively. OS at five and 10 years varied according to the number of TKIs received as follows: one TKI (*n* = 76), 100% OS at both five and 10 years; two TKIs (*n* = 73), 95% at five years and 82% at 10 years; three TKIs (*n* = 33), 97% at five years and 87% at 10 years; four TKIs (*n* = 16), 75% at both five and 10 years; five TKIs (*n* = 7), 82% at five years and 55% at 10 years (Figure 1b, *p* = 0.002). The best OS rate was observed in first-line TKI-treated patients.

### 3.3. Outcomes According to First-Line TKI: IM vs. NG-TKI

After diagnosis, 70.4% (*n* = 145) of patients started IM and 29.6% (*n* = 61) were treated with NG-TKI (Table 2). Among the NG-TKI group, 30 patients changed to IM and six to another NG-TKI. The incidence of moving to a second-line TKI treatment was not different between the two groups (*p* = 0.639). The median follow-up period of the IM group was longer than that of the NG-TKI group (80.9 vs. 23.6 months, *p* = 0.000). The quantitative RT-PCR results for *BCR/ABL1* (BCR/ABL1^IS^) at three, six, and 12 months were available for 110 (53.9%), 106 (51.5%) and 101 (49%) patients, respectively. The reasons why approximately half of the quantitative RT-PCR results were missing was that quantitative RT-PCR was performed from 2010, and it was not performed in a local clinic because most patients were referred from the local clinic. Based on the available early molecular response, more patients within the NG-TKI group achieved BCR/ABL1^IS^ ≤10% at three months (*p* = 0.003), <1% at six months (*p* = 0.012), and ≤0.1% at one year (*p* = 0.013) compared to the IM group. In the late molecular evaluation, it was discovered that more NG-TKI-treated patients maintained “BCR/ABL1^IS^ <0.1” than IM-treated patients without statistical significance (*p* = 0.086). The incidence of disease progression and death were not different between the two groups (*p* = 0.066 and *p* = 0.727) (Table 2).

### 3.4. Factors Affecting PFS or OS

Univariate analysis was performed with the variables described in the methods above. Female patients showed better PFS than male (*p* = 0.009). Patients achieving 1 log, 2 log, and 3 log *BCR/ABL* reduction at three, six and 12 months showed a longer PFS than those who did not (*p* < 0.001, *p* = 0.002, and 0.037, respectively) (Figure 2a,c,e). Those who developed *ABL* kinase mutations showed a significantly decreased PFS than those without the mutation (*p* < 0.001).

Hasford (high vs. intermediate and low risk, *p* < 0.001) and EUTOS (high vs. low, *p* = 0.047) scores could predict OS. *ABL* kinase mutated patients had poorer OS than those without *ABL* kinase mutations or patients not tested (*p* < 0.001). Patients with BCR/ABL1^IS^ ≤10% at three months and <1% at six months showed better OS than those with BCR/ABL1^IS^ >10% at three months (*p* = 0.002) and ≥1% at six months (*p* = 0.013); however, optimal responses at 12 months did not affect OS (*p* = 0.112) (Figure 2b,d,f). Lastly, age (<50 vs. ≥50), race, Sokal risk score, additional chromosomal abnormality, and first-line TKI (IM vs. NG-TKI) did not affect either PFS or OS.

We analyzed the OS of 130 patients according to reason for switch from the 1st TKI, resistance (*n* = 66), intolerance (*n* = 50) and other reasons (*n* = 14), but it was not significant (data was not shown).

### 3.5. TKI Discontinuation

A total 124 patients (60.2%) achieved complete molecular response (CMR) at least once during the follow up period (Table 3). With the first-line TKI, 69 patients (33.5%) achieved CMR. Additionally, 41, eight, five and one patients experienced CMR during treatment with a third-, fourth- and fifth-line TKI, respectively, who never achieved CMR in prior treatment (Table 3). Fourteen patients attempted TKI discontinuation at first-line TKI, all patients were taking IM, and half of them (*n* = 7) ceased medication. During the second-line treatment, seven stopped TKIs (DAS, *n* = 5; NIL, *n* = 1; BOS, *n* = 1). Two of them had achieved CMR from the first-line treatment, but five achieved CMR after second-line treatment. One patient who tried to stop third-line TKI (BOS) achieved CMR since the second-line treatment. At fourth-line treatment, three discontinued IM (*n* = 2) and PON (*n* = 1), and they achieved deep molecular response at second-, third- and fourth-line treatments, respectively. TKIs were restarted at a median of 2.7 months (range, 2.7–24.5) after TKI discontinuation due to loss of MMR (*n* = 10) and TKI withdrawal symptoms (*n* = 1). Thus, treating CP CML with more than one TKI enabled an additional 11 patients (5.3%) to attempt discontinuing TKIs. Overall, 25 patients (12.1%) stopped TKIs for a median of 42.3 months (range, 15.4–154.3) after achieving CMR, and 14 (6.8%) stopped TKIs for a median of 6.3 months (range 1–53.4).

## 4. Discussion

In this retrospective study, we evaluated the real-world practice of treating CP CML in the era of multiple TKIs and analyzed the long-term outcomes and any associated factors. Among the 206 patients, the estimated 10-year PFS and OS of CP CML patients (*n* = 206) were 81% and 87%, respectively. Specifically, the best OS was observed in patients tolerating and responding to first-line TKI treatment, and the worst was observed in those treated with 5 TKIs (Figure 1b). The outcome of first-line TKI treatment was compatible with previous reports [21,22,23]. Beyond first-line TKI treatment, to the best of our knowledge, there were no available data for comparison with outcomes of 1–5 TKIs for this study. Akosile et al. reported the OS of CP CML, which was measured from the second-line TKI, and the five-year OS of patients who used two, three and four or more TKIs were 80%, 53% and 38%, respectively [24]. The OS was not measured from the diagnosis, but this report showed similar trends with our study indicating that the best OS was observed in the 2-TKI group with statistical significance (*p* < 0.001) [24].

As the life expectancy of the newly diagnosed CP CML is now very close to that of age matched individuals in the general population [2], the aim of the treatment is now moving to treatment-free remission (TFR) in order to improve the quality of life and avoid long-term toxicities [15]. This study revealed that multiple TKI treatment enabled patients who could not achieve a CMR with frontline TKI to achieve deep molecular response and give them opportunities to stop their TKI treatments. We reported the clinical course of patients who discontinued TKIs in our institution [25], which included 16 patients who had attempted TKI discontinuation in this study. Indeed, multiple TKI improved not only OS but also TFR.

Although the worst outcome was observed in those treated with five TKIs, the OS of patients beyond first-line TKI treatment was still high, thanks to newer generation TKIs. Second generation (2G) TKIs demonstrated faster and higher rates of response compared with IM [21,22]. However, IM was the most commonly used first-line TKI treatment. Even after the FDA approval of DAS and NIL for treatments for adults with newly diagnosed CML (DAS and NIL were approved in October 2010 and June 2010, respectively), IM was not fully replaced by DAS or NIL. One hundred thirteen patients started TKI treatment after July 2010, including five that were treated with PON and BOS. IM was still the most frequently selected TKI by physicians (*n* = 52, 46%), even though either as a frontline or second-line treatment, DAS was preferred to NIL (DAS *n* = 43, 38%; NIL, *n* = 13, 12%), and this is compatible with previous reports [7,26].

In this study, with multiple TKI-treated patients, Hasford and EUTOS had a prognostic impact on OS, but the Sokal score did not. According to the ELN guidelines, these three prognostic score systems still have value, and there was no evidence that any one of the three risk scores was superior or more convenient [18]. However, this was based on the results of frontline IM treatment [18]. There are studies reporting that EUTOS did not predict the survival of CP CML patients treated with frontline 2G TKI [27,28,29]. Thus, we need to validate prognostic score systems in multiple TKI-treated patients.

In recent years, patients reaching molecular milestones at three, six and 12 months have shown better event-free survival (EFS), PFS or OS, and were either treated with frontline IM or 2G TKIs [30,31,32,33,34,35]. In this study, including multiple TKI-treated patients, molecular response at three, six and 12 months showed a prognostic value. Notably, achieving 1 log and 2 log reductions of *BCR/ABL1* transcripts at three and six months predicted favorable outcomes (for both PFS and OS). Interestingly, more frontline NG-TKI-treated patients reached molecular milestones at three, six and 12 months than frontline IM-treated patients (Table 2); however, this did not translate to PFS or OS, which was consistent with previous reports [21,22].

One of the interesting findings of this study is that the female sex was associated with better PFS than the male sex (*p* = 0.009). There are several reports about sex and the outcomes of CML; female patients had lower hazard ratios of death than male patients in a 5784-patient study [36], and this strongly predicted stable undetectable *BCR-ABL1* in IM-treated CP CML male patients [30]. Moreover, female patients had higher rates of adherence to IM than male patients [37]. However, there was no report about PFS and sex; thus, further investigation is warranted.

In conclusion, among 206 CP CML patients, 63% were treated with more than one TKI. Improved OS was observed in patients tolerating and responding to the first-line TKI treatment. Although the number of TKIs used differentiated the OS, the OS beyond first-line TKI treatment was favorable and multiple TKIs enabled more patients to achieve TFR. Early molecular response at three and six months and the Hasford or EUTOS risk scores were still valuable in predicting the OS.

## Figures and Tables

**Figure 1 jcm-09-01542-f001:**
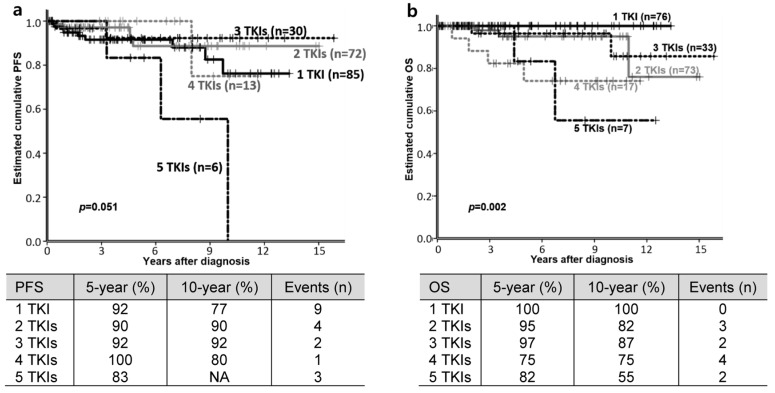
Estimated cumulative progression free survival (PFS) (**a**) and overall survival (OS) rate (**b**) of chronic phase chronic myeloid leukemia patients according to the number of treated tyrosine kinase inhibitors (TKIs). (Since 15 patients were treated with a median of 1.5 (1–3) or more TKIs after progression, patient numbers of the 1–5 TKI-treated group in PFS were different from OS.).

**Figure 2 jcm-09-01542-f002:**
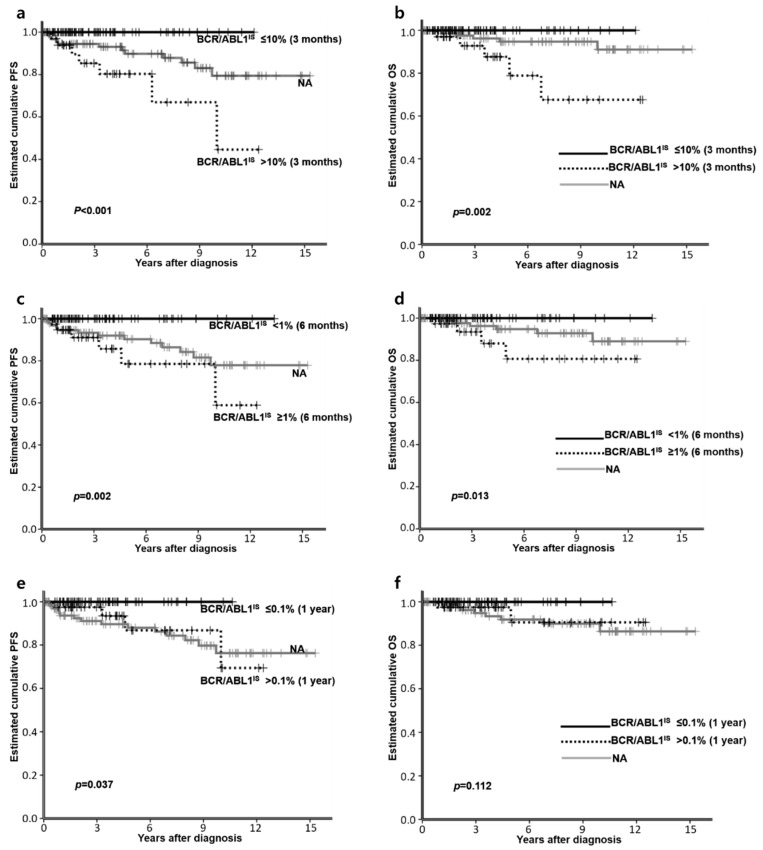
Event free survival (PFS) and overall survival (OS) depending on BCR/ABL1IS at 3 months (**a**,**b**), 6 months (**c**,**d**), and 12 months (**e**,**f**). All *p* values are calculated using the log-rank test with available data only.

**Table 1 jcm-09-01542-t001:** Patients’ clinical characteristics.

		*n*	%
Median Age at Diagnosis, Years (Range)		^a^ 50 (11–88)	
Sex	M/F	105/101	51/49
Race	African American	56	27.2
	Asian	5	2.4
	Hispanic	6	2.9
	White	106	51.5
	Others	3	1.5
	Unknown or declined	30	14.6
Risk (available, *n* = 110)	Sokal Low/Intermediate/High	64/27/19	58.2/24.5/17.3
	Hasford Low/Intermediate/High	66/38/6	60.0/34.5/5.5
	EUTOS Low/High	95/15	86.4/13.6
^b^ Additional chromosomal abnormality (available, *n* = 173)		12	6.9
Number of used TKI	1	76	36.9
	2	73	35.4
	3	33	16.0
	4	17	8.3
	5	7	3.4
1st-line TKI (*n* = 206)	IM/DAS/NIL/BOS/PON	145/43/13/4/1	70.4/20.9/6.3/1.9/0.5
2nd-line TKI (*n* = 130)	IM/DAS/NIL/BOS/PON/REB	28/68/23/9/1/1	21.5/52.3/17.7/6.9/0.8/0.8
3rd-line TKI (*n* = 65)	IM/DAS/NIL/BOS/PON	8/19/21/11/6	12.3/29.2/32.3/16.9/9.2
4th-line TKI (*n* = 30)	IM/DAS/NIL/BOS/PON/REB	4/4/1/9/11/1	13.3/13.3/3.3/30.0/36.7/3.3
5th-line TKI (*n* = 10)	DAS/BOS/PON	3/3/4	30.0/30.0/40.0
Reason for TKI switch or discontinuation (239 events)	R	^c^ 119	49.8
	I	^c^ 104	43.5
	O	^c^ 16	6.7
Nivolumab		3	1.5
Blinatumomab		1	0.5
Chemotherapy		12	5.8
HSCT	Allogeneic/Autologous	8/1	3.9/0.5
^d^*ABL* domain mutation (*n* = 72)	Positive	25	34.7

Abbreviations: TKI, tyrosine kinase inhibitor; IM, Imatinib: DAS, Dasatinib: NIL, Nilotinib: BOS, Bosutinib: PON, Ponatinib: REB, Rebastinib; R, resistance, suboptimal response or treatment failure; I, intolerance or adverse events; O, others (insurance, clinical trial, or not available); HSCT, hematopoietic stem cell transplantation ^a^ median years (range). ^b^ Additional chromosomal abnormality: add(3)(q21); −6; del(7)(q22q32); +8; del(10)(q24); del(13)(q12q22); add(15)(q26.1); del(16)(q22); +19; +21; t(X;10)(q13;q26); +Y; der(5)t(1;5)(q21;q23), add(9)(q32), der(9), del(9)(q34), der19(q13.2), t(5;22)(q33;q11) ^c^ “event” instead of “*n*” ^d^
*ABL* domain mutation; E255K, E355A, E459K, F317L, E317V, E453K, F359C, F359V, G250E, L248V, M244T, M351T, Y253F, Y253H, T315I.

**Table 2 jcm-09-01542-t002:** Outcomes according to the front line tyrosine kinase inhibitor, Imatinib vs. newer generation tyrosine kinase inhibitors.

*n* (%)		IM (*n* = 145)	NG-TKI (*n* = 61)	*p*
**Switching TKIs**		93 (64.1)	37 (60.7)	0.639
Median follow up, months (range)		80.9 (1.4–190.1)	23.6 (1.7–64.6)	0.000
BCR/ABL1^IS^ at 3 months (*n* = 111)	≤10%	39 (59.1)	39 (86.7)	0.003
	>10%	27 (40.9)	6 (13.3)	
BCR/ABL1^IS^ at 6 months (*n* = 106)	<1%	36 (54.5)	32 (80.0)	0.012
	≥1%	30 (45.5)	8 (20.0)	
BCR/ABL1 at 1 year (*n* = 103)	≤0.1%	33 (51.6)	30 (76.9)	0.013
	>0.1%	31 (48.4)	9 (23.1)	
The last BCR/ABL1^IS^ <0.1		83 (57.2)	43 (70.5)	0.086
Disease progression		17 (11.7)	2 (3.3)	0.066
Death		9 (6.2)	2 (3.3)	0.513

Abbreviations: TKI, tyrosine kinase inhibitor; IM, Imatinib; NG-TKI, newer generation tyrosine kinase inhibitor.

**Table 3 jcm-09-01542-t003:** Percentages of complete molecular response and TKI discontinuation at each treatment line.

TKI	^1^ CMR, *n* (%)	^2^ Additional CMR, *n* (%)	^3^ Attempts of TKI Discontinuation, *n* (%)	^4^ TKI Discontinuation, *n* (%)
1st-line TKI (*n* = 206)	69 (33.5)	-	14 (6.8)	7 (3.4)
2nd-line TKI (*n* = 130)	52 (40.0)	41 (31.5)	7 (5.4)	5 (3.8)
3rd-line TKI (*n* = 65)	18 (27.7)	8 (12.3)	1 (1.5)	1 (1.5)
4th-line TKI (*n* = 30)	8 (26.7)	5 (16.7)	3 (10.0)	1 (3.3)
5th-line TKI (*n* = 10)	1 (10.0)	1	0	0 (0)
Total (*n* = 206)	124(60.2)	-	25(12.1)	14 (6.8)

^1^ Number of patients who achieved CMR at least once. ^2^ Patients who never achieved CMR and achieved it after switching TKIs. ^3^ Attempts of TKI discontinuation not because of intolerance or non-adherence, but with the purpose of permanent discontinuation. ^4^ Patients who stopped TKI at the last follow up without loss of MMR. **Abbreviations:** TKI, tyrosine kinase inhibitor; CMR, complete molecular response; MMR, major molecular response.

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
