# Peer review of "Outcomes of Chronic Phase Chronic Myeloid Leukemia after Treatment with Multiple Tyrosine Kinase Inhibitors"

_jcm, 2020, doi:10.3390/jcm9051542_

Round 1

Reviewer 1 Report

The paper entitled “Outcomes of chronic phase chronic myeloid leukemia after treatment with multiple tyrosine kinase inhibitors” wrote by Kong and colleagues reported the outcomes of 206 CML patients in chronic phase treated with 5 different tyrosine kinase inhibitors (TKIs) in 1st, 2nd, 3rd, 4th or 5th lines. The paper is well written but I have some doubts about the number of samples and the subsequent statistical analysis. In my opinion a study of Overall Surviaval (OS) in a cohort where happened only 11 events (5%) with 4 deaths due to complications of allogeneic HSCT, it needs more than 200 pts to obtain robust KM curves for every variable (type of TKI, number of treatments, etc). For instance, Akosile et al. in an ASH poster performed a very similar analysis using 1775 pts to obtain good KM curves. Despite these considerations, the authors demonstrated that the use of multi TKIs could be not so toxic in CML treatment

My comments are:

  • My first comment is about the principal results. In my opinion the aim of a study like yours is to understand if a patient with resistance or intolerance to TKI have the potential for long-term survival and if it is better to treat him 3, 4 or 5 times or it is better to stop or change treatment. In your results you affirmed “The OS rate of 1-TKI treatment was different from that of 2–5-TKI treatments”. This is an obvious result because who switch to an additional treatment is probably resistant to standard TKI treatment. Please make the same analysis comparing each other groups of patients treated 2,3,4 or 5 times.
  • Unfortunately your database lacks of several information about some clinical and molecular data. You have got full data for only 50% of patients. In this context you analyzed factors affecting PFS or OS in the whole cohort and independently of number of TKI used. It is possible to make analysis of factors in relation to number of TKIs or number of treatments?
  • In last years several papers discussed about discontinuation of TKIs. In your dataset can you prove that several treatments with TKIs give better OS and PFS respect discontinuation?
  • Have you got some data about CML patients treated by other treatments than TKI? OS and PFS are better in TKI treated patients?
  • why in the results (line 96) you affirmed “A total of 206 patients, with a median follow-up period of 48.8 (1.4–190.1) months after diagnosis” and later (line 141) “At a median follow-up period of 39.0 (3.8-119.8) months after diagnosis”. Are not the same patients? Why follow-ups are different?
  • TKI 1st line treatments were performed early after diagnosis? Maybe is it better to use follow up after treatment to compare better effect of multi TKIs?

Author Response

Thank you for giving me the opportunity to submit a revised draft to Journal of Clinical Medicine. We appreciate the time and effort that you and the reviewers have dedicated to providing your valuable feedback on my manuscript. We have been able to incorporate changes to reflect most of the suggestions provided by the reviewers. We have highlighted the changes within the manuscript. Here is a point-by-point response to the reviewers’ comments and concerns.

  1. Unfortunately your database lacks of several information about some clinical and molecular data. You have got full data for only 50% of patients. In this context you analyzed factors affecting PFS or OS in the whole cohort and independently of number of TKI used. It is possible to make analysis of factors in relation to number of TKIs or number of treatments?

 Age at diagnosis, sex, type of 1st line TKI were not associated with number of TKIs, but patients who achieved early molecular response at 3 months tended to maintain their 1st line TKI. We agree that this is does not match the flow of contents, and therefore we did not mention it in the manuscript.

  1. In last years several papers discussed about discontinuation of TKIs. In your dataset can you prove that several treatments with TKIs give better OS and PFS respect discontinuation?

We totally agree that we need to discuss the topic of TKI discontinuation. We added to it section 3.5 and revised the discussion section.

  1. Have you got some data about CML patients treated by other treatments than TKI? OS and PFS are better in TKI treated patients?

Thank you for your comment. I am not completely sure what exactly ‘other treatments than TKI’ means, but if it is chemotherapy or interferon, then we do not have that data.

  1. why in the results (line 96) you affirmed “A total of 206 patients, with a median follow-up period of 48.8 (1.4–190.1) months after diagnosis” and later (line 141) “At a median follow-up period of 39.0 (3.8-119.8) months after diagnosis”. Are not the same patients? Why follow-ups are different?

I revised the sentence to prevent confusion.

“At a median follow-up period of 39.0 (3.8-119.8) months after diagnosis  -> At median 39.0 (3.8-119.8) months after diagnosis… “

  1. TKI 1st line treatments were performed early after diagnosis? Maybe is it better to use follow up after treatment to compare better effect of multi TKIs?

Median time from diagnosis to TKI initiation was 0.3 (0-13.8) months and only 4 patients started TKI 3 months after diagnosis. Moreover, the TKI start date was missing in one patient. The outcome did not differ when we analyzed with the date of TKI initiation in 205 patients.

Reviewer 2 Report

The authors evaluated the outcomes of chronic myeloid leukemia patients (chronic phase) treated with the 5 tyrosine kinase inhibitors that are commercially available for this hematological neoplasm i.e. Imatinib, dasatinib, nilotinib, bosutinib and ponatinib. They have a series of limitations, such as the unavailability of the qRT-PCR technique in cases diagnosed before 2010. Despite this, they are able to draw conclusions, in terms of PFS and OS, related to the molecular response at 3, 6 and 12 months), consistent with the published data, which may it validates in some way their cohort.

Comments:

  1. The introduction provides a generalized background of the topic and the authors provides references to other groups who do or have done research in this area. On the other hand the motivations, as well as the objective for this study is clear defined.

  1. In general, the methods part, particularly the statistical part, should be more detailed.

pag.2; 83-84

The authors said “the Mann-Whitney U test and ANOVA were used for continuous variables”.

When they use Mann-Whitney U test, I believed that the authors have tested for normality. Following the same criteria, when they use one way ANOVA (I supposed they refers to this test), first the authors should be test for normality and homogeneity assumptions. If the data do not meet one of these two assumptions or both, a non-parametric test must be taken, i.e. Kruskal-Wallis.  Although I don't see where they could have used the ANOVA test.

pag.2; 84-85

The authors must defined progression-free survival. They said “Progression-free survival (PFS) and overall survival (OS) rates were calculated from date of diagnosis to date of progression, death, or last follow-up using the life tables and Kaplan-Meier method”, but they did not define what the event of progression is, it is mean progression to accelerated or blastic phase or to death?

  1. According with the results section, the way the author represent the data are clear, however there are many new results data in discussion that should better describe how to obtain them.

How the authors extract multiples p-value from Kaplan-Meier curves. The log-rank test report only one p-value, but doesn't tease out which curves are different. One of multiples examples: when the authors said “The OS rate of 1-TKI treatment was different from that of 2–5-TKI treatments (except 3-TKI treatment) (Figure 1B).” in pag. 7; 200-201.

In addition, it would be interesting, even if it were in supplementary data, some descriptive regarding the ABL TK mutations found, not a simple enumeration like the one shown in Table 1. In this piece of information it could be add what treatment the patient had before and after find the mutation.

Author Response

Thank you for giving me the opportunity to submit a revised draft to Journal of Clinical Medicine. We appreciate the time and effort that you and the reviewers have dedicated to providing your valuable feedback on my manuscript. We have been able to incorporate changes to reflect most of the suggestions provided by the reviewers. We have highlighted the changes within the manuscript. Here is a point-by-point response to the reviewers’ comments and concerns.

1. The introduction provides a generalized background of the topic and the authors provides references to other groups who do or have done research in this area. On the other hand the motivations, as well as the objective for this study is clear defined.

We added TFR according to the comments of reviewer 1 and 2.

2. In general, the methods part, particularly the statistical part, should be more detailed.

We reviewed all statistics with statistician, found some errors you pointed out. We expressed thanks to her in acknowledgments. 

pag.2; 83-84

The authors said “the Mann-Whitney U test and ANOVA were used for continuous variables”.

When they use Mann-Whitney U test, I believed that the authors have tested for normality. Following the same criteria, when they use one way ANOVA (I supposed they refers to this test), first the authors should be test for normality and homogeneity assumptions. If the data do not meet one of these two assumptions or both, a non-parametric test must be taken, i.e. Kruskal-Wallis.  Although I don't see where they could have used the ANOVA test.

We did not perform an ANOVA analysis, and removed “and ANOVA were…”. 

pag.2; 84-85

The authors must defined progression-free survival. They said “Progression-free survival (PFS) and overall survival (OS) rates were calculated from date of diagnosis to date of progression, death, or last follow-up using the life tables and Kaplan-Meier method”, but they did not define what the event of progression is, it is mean progression to accelerated or blastic phase or to death?

We revised “Progression-free survival (PFS) and overall survival (OS) rates were calculated from date of diagnosis to date of progression, death, or last follow-up using the life tables and Kaplan-Meier method” to “. Progression-free survival (PFS) was calculated from date of diagnosis to date of accelerated phase or blastic crisis, and overall survival (OS) rates were calculated from diagnosis to death, or last follow-up using the life tables and Kaplan-Meier method.”

3. According with the results section, the way the author represent the data are clear, however there are many new results data in discussion that should better describe how to obtain them.

How the authors extract multiples p-value from Kaplan-Meier curves. The log-rank test report only one p-value, but doesn't tease out which curves are different. One of multiples examples: when the authors said “The OS rate of 1-TKI treatment was different from that of 2–5-TKI treatments (except 3-TKI treatment) (Figure 1B).” in pag. 7; 200-201.

For the comparison of each of the 5 groups (OS of 1-TKI vs 2-TKI, 3-TKI, 4TKI, and 5-TKI… and so on), we should use another statistical method. Thus, we removed “The OS rate 1-TKI treated group was better than that of the 2-TKI (p-0.046), 4-TKI (P<0.001), or 5-TKI (p<0.001) treated groups but it was not different from the 3-TKI treated group (p=0.007)” in the end of paragraph or ‘result 3.2’, and “The OS rate of 1-TKI treatment was different from that of 2–5-TKI treatments (except 3-TKI treatment) (Figure 1B).” in the first paragraph of the discussion.

In addition, it would be interesting, even if it were in supplementary data, some descriptive regarding the ABL TK mutations found, not a simple enumeration like the one shown in Table 1. In this piece of information it could be add what treatment the patient had before and after find the mutation.

Thank you for your comments. Because mutation and additional chromosomal analysis was done at different time points and the treatment was very different from one another (and some dates when the mutation test was performed were missing), we could not perform a statistical analysis. Thus we have just included a description in the manuscript.